# L1: Controlling How Long A Reasoning Model Thinks With Reinforcement Learning

**Pranjal Aggarwal**
Carnegie Mellon University

**Sean Welleck**
Carnegie Mellon University
{pranjala,swelleck}@cs.cmu.edu

## Abstract

Reasoning language models have shown an uncanny ability to improve performance at test-time by "thinking longer"—that is, by generating longer chain-of-thought sequences and hence using more compute. However, the length of their chain-of-thought reasoning is not controllable, making it impossible to allocate test-time compute to achieve a desired level of performance. We introduce Length Controlled Policy Optimization (LCPO), a simple reinforcement learning method that optimizes for accuracy and adherence to user-specified length constraints. We use LCPO to train L1, a reasoning language model that produces outputs satisfying a length constraint given in its prompt. L1's length control allows for smoothly trading off computational cost and accuracy on a wide range of tasks, and outperforms the state-of-the-art S1 method for length control. Furthermore, we uncover an unexpected short chain-of-thought capability in models trained with LCPO. Specifically, using LCPO we derive Short Reasoning Models (SRMs), that exhibit similar reasoning patterns as full-length reasoning models, but can generate CoT lengths comparable to non-reasoning models. They demonstrate significant performance gains, for instance, our 1.5B L1 model surpasses GPT-4o at equal reasoning lengths. Overall, LCPO enables precise control over reasoning length, allowing for fine-grained allocation of test-time compute and accuracy. [1]

## 1  Introduction

An emerging class of *reasoning language models* (OpenAI et al., 2024a; DeepSeek-AI et al., 2025) improve performance at test-time by thinking longer when solving complex problems—that is, by generating extended chain-of-thought (Wei et al., 2023) sequences and in turn using more compute. However, current reasoning models have a key limitation: the length of their reasoning is uncontrolled, making it impossible to allocate a test-time compute budget to achieve a target performance level. In some cases, sequences span tens of thousands of tokens, wasting compute, while in others, models stop too early on complex problems.

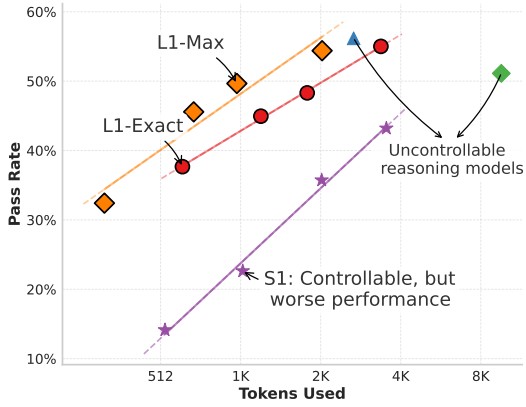

Figure 1: L1 is a length-controllable reasoning model, that achieves higher performance per token than other methods.

Recent methods like S1 (Muennighoff et al., 2025) attempt to achieve length control by forcing a model to generate special tokens (e.g., "Wait", "Final Answer") when the generation is too short or too long. However, this rigid, hand-engineered strategy severely

---

[1]Code and models released at https://cmu-l3.github.io/l1

degrades performance compared to the underlying model (Figure 1). Other work investigates controlling output lengths in instruction following and general domains (Butcher et al., 2024; Yuan et al., 2024). However, reasoning models face fundamentally new challenges such as much longer output lengths and the need to trade off computational cost for improved performance.

We propose **Length Controlled Policy Optimization (LCPO)**, a simple reinforcement learning (RL)-based method that gives reasoning language models precise and adaptive length control. LCPO trains models to satisfy two objectives: (1) correctness of the final output, and (2) generating reasoning sequences that meet a length constraint specified in the prompt. In doing so, LCPO-trained models *learn* to satisfy length constraints while optimizing reasoning performance rather than relying on hand-engineered heuristics.

We experiment with two practical constraints: (1) LCPO-Exact, which requires the generated reasoning to be exactly equal to the target length, and (2) LCPO-Max, which requires the output to be no longer than the target length. We use LCPO to fine-tune a 1.5B-parameter reasoning model based on Qwen-Distilled-R1-1.5B (DeepSeek-AI et al., 2025; Qwen et al., 2025), producing L1-Max and L1-Exact. Our L1 models can precisely trade off token budget and reasoning performance, smoothly interpolating between short, efficient reasoning and longer, more accurate reasoning by simply prompting the model with different length constraints (Figure 1). Crucially, one point on this trade-off curve recovers the original base model's performance, while outperforming S1 (Muennighoff et al., 2025) in performance across the entire range of reasoning lengths (Figure 1). On math reasoning tasks, L1 outperforms S1 by up to 100% relative and 20% absolute, under identical conditions.

Beyond improved length control in the standard math reasoning setting, we find that LCPO-trained models generalize surprisingly well to out-of-distribution tasks, including logical reasoning, and general-knowledge benchmarks like MMLU (Hendrycks et al., 2021a). Furthermore, we show that "long-CoT" models trained with LCPO become unexpectedly strong "short-CoT" models: when prompted to generate short reasoning traces, LCPO-trained models outperform their original counterparts by significant margins (up to 10% improvement), even at the same generation length. We refer these models as Short Reasoning Models (SRMs). To the best of our knowledge, for the first time we show that a 1.5B model can match the performance of GPT-4o (OpenAI et al., 2024b) despite using the same token budget. In summary, our contributions are:

- We introduce Length Controlled Policy Optimization (LCPO), the first reinforcement learning-based method for training reasoning language models that produce outputs adhering to user-specified length constraints.

- We use LCPO to train L1, which demonstrates high degree of length control and achieves state-of-the-art reasoning accuracy at fixed token budgets on challenging math reasoning benchmarks.

- We show length-control of L1 generalizes beyond math reasoning tasks to diverse out-of-distribution tasks, including logical reasoning, and general-domain benchmarks (MMLU).

- We demonstrate that LCPO-trained models can act as strong short-CoT models, significantly outperforming their non-reasoning counterparts and much larger models such as GPT-4o, despite using the same token budget.

## 2 Related Work

**Test-Time Scaling in Large Language Models.** Increasing test-time computation has consistently been shown to improve performance in complex reasoning tasks, mathematical problem-solving, and code generation (Wu et al., 2024; Wang et al., 2023; Wei et al., 2023; DeepSeek-AI et al., 2025; Snell et al., 2024). Test-time scaling laws indicate predictable performance gains from increasing inference computation, either by generating more reasoning chains or longer ones (Wu et al., 2024; Snell et al., 2024; OpenAI et al., 2024a). Prominent approaches include parallel sampling of multiple reasoning paths (Wang et al., 2023; Aggarwal et al., 2023), tree-based search (Yao et al., 2023; Wu et al., 2024; Xin et al., 2024), and iterative refinement techniques (Welleck et al., 2023; Madaan et al., 2023; Snell et al.,

2024; Welleck et al., 2024). Recent reasoning language models such as "O1" and "R1"-style models (OpenAI et al., 2024a; DeepSeek-AI et al., 2025) simplify test-time scaling by generating extended reasoning traces (longer chains-of-thought). Despite their promising results, these methods lack precise and dynamic control over the length of the generated reasoning chains, resulting in often suboptimal performance or unrealized potential efficiency gains. Our work complements and extends this line of research by enabling reasoning models to precisely control the length of generated outputs, thereby providing flexibility to calibrate inference compute based on task-specific requirements.

**Length Control in Large Language Models.** Controlling the length of LLM-generated outputs is an important practical consideration across various generation tasks. Approaches proposed thus far include architectural modifications—such as manipulating positional encodings for exact sequence-length generation (Butcher et al., 2024)—training objective adjustments to explicitly enforce length constraints (Jie et al., 2023; Singhal et al., 2024), or directly training models on instruction-style data explicitly labeled with desired output lengths (Yuan et al., 2024). Previous works on length control largely fall into two use-case categories. The first aims primarily to reduce unnecessary verbosity (as often desired in RLHF-tuned instruction-following models) while the second aims either to impose maximum length budgets or achieve precise token-level length adherence (Jie et al., 2023; Yuan et al., 2024; Singhal et al., 2024). However, existing methods predominantly focus on general-purpose text generation or instruction-following contexts, where cost-quality efficiency trade-offs are less critical or remain unaddressed (Jie et al., 2023; Yuan et al., 2024; Butcher et al., 2024). Our work addresses the new challenges present in reasoning models.

Length control specifically tailored to reasoning tasks remains relatively unexplored. Prior works have shown that reasoning models often underthink or overthink, both hurting performance and wasting compute (Wang et al., 2025; Chen et al., 2025). Recent works, such as those by Arora & Zanette (2025) and Kang et al. (2024), emphasize generating shorter reasoning chains for efficiency, but they do not enable explicit length control or precise alignment with user-specified inference budgets. Another work S1 (Muennighoff et al., 2025) introduces "budget-forcing" by imposing a strict token limit: either truncating output at budget exhaustion or inserting a special token ("Wait") to request continued generation until reaching the full length budget. Unfortunately, this strategy presents significant practical drawbacks. Abrupt truncation often interrupts reasoning mid-step, negatively impacting model accuracy and user interpretability. Meanwhile, the repetitive usage of special continuation tokens risks rigid and suboptimal reasoning patterns.

In contrast to these prior works, LCPO is uniquely designed to train reasoning-specialized models for precise and adaptive length control. LCPO uses reinforcement learning so that models *learn* to dynamically allocate inference compute based on constraints provided in a prompt. As our experiments will demonstrate, our method substantially surpasses previous approaches in precision over length control, and performance at varying length budgets.

## 3 Method

Current reasoning language models lack an explicit mechanism for controlling the length of their generated reasoning traces. This limitation prevents users and downstream applications from explicitly calibrating the inference compute budget (number of generated tokens) according to task-specific requirements or available computational resources.

In this work, we address this limitation by conditioning the model on a target token length provided in the prompt. Formally, given an input prompt $x$ and a target length $n_{gold}$, the model is expected to generate a response $y$ whose length $n_y$ minimizes the absolute difference $|n_{gold} - n_y|$ while simultaneously producing the correct answer. This formulation directly couples accuracy with output length, ensuring that the generated chain-of-thoughts adhere to user-specified constraints.

**Length Controlled Policy Optimization.** We begin with a pre-trained reasoning language model $LLM_\theta$ and a dataset $D = \{(x_i, y_{gold,i})\}_{i=1}^{N}$, where each instance contains only the

input prompt and the final answer (i.e., no intermediate reasoning traces). To enable length control, each prompt $x_i$ is augmented by appending a target length instruction. In particular, we form

$$x_i^{new} = \text{Concat}\Big(x_i, \text{``Think for } n_{gold,i} \text{ tokens.''}\Big),$$

where $n_{gold,i}$ is sampled uniformly from $\mathbb{Z}(n_{min}, n_{max})$. This augmentation yields a new dataset $D^{new} = \{(x_i^{new}, y_{gold,i})\}_{i=1}^N$.

We then update $LLM_\theta$ using a reinforcement learning objective. In our experiments we adopt GRPO (Shao et al., 2024) (though the method is compatible with other RL algorithms). Our reward function combines two terms: a correctness reward $r_c$ and a length penalty $r_{length}$. It is defined as

$$r(y, y_{gold}, n_{gold}) = \mathbb{I}(y = y_{gold}) - \alpha \cdot \big|n_{gold} - n_y\big|, \tag{1}$$

where $\mathbb{I}(\cdot)$ is the indicator function, $n_y$ is the generated output length, and $\alpha$ is a scalar that regulates the trade-off between generating the correct answer and meeting the target length. In practice, a lower value of $\alpha$ prioritizes correctness when it is critical, whereas a higher value enforces stricter adherence to the length constraint. Notably, the reward function serves a dual purpose: (a) it encourages the model to produce correct answers while implicitly favoring concise reasoning traces when shorter outputs are requested, and (b) it consistently motivates the model to match the prescribed target length even when a correct answer could be generated with fewer tokens. We refer to the model trained with this objective as L1-Exact.

At inference, the output length is controlled by selecting a fixed target length $n_{gold}$ (or a set of lengths) that is appended uniformly to every test prompt.

**Maximum Length Constraint Mode.** We further train a variant of L1 called L1-Max, which flexibly generates outputs of varying lengths while respecting a maximum length constraint. This approach is valuable when users prioritize staying within a computational budget rather than adhering to exact generation lengths. To train L1-Max, we fine-tune the L1-Exact model using the same RL framework but with a modified reward function:

$$r(y, y_{gold}, n_{gold}) = \mathbb{I}(y = y_{gold}) \cdot \text{clip}(\alpha \cdot (n_{gold} - n_y) + \delta, 0, 1), \tag{2}$$

where $\alpha$ controls the penalty for length violations. This formulation applies a soft constraint that (1) gradually penalizes outputs exceeding the target length rather than imposing a hard cutoff (which is necessary to ensure gradient propagation in GRPO objective), and (2) incentivizes the model to use fewer tokens when possible without sacrificing correctness. The $\delta = 0.5$ term ensures that correct answers with minor budget violations are still preferred over incorrect answers. Further, L1-Max is trained with dual objective: when the prompt requests an exact length, the model uses Equation 1; otherwise, it defaults to the maximum constraint mode using Equation 2.

## 4 Experimental Setup

**Models and Datasets.** We conduct training on the DeepScaleR-Preview-Dataset (Luo et al., 2025), a mathematics dataset consisting of 40K question-answer pairs drawn from AIME, AMC, Omni-Math (Gao et al., 2024) and STILL (Min et al., 2024). We evaluate our models on test sets of 4 different reasoning datasets: AIME 2025, MATH (Hendrycks et al., 2021b), AMC, Olympiad-Bench (He et al., 2024), and additionally GPQA (Rein et al., 2023), LSAT (Zhong et al., 2023), and MMLU (Hendrycks et al., 2021a). Our base model is DeepScaleR-1.5B-Preview, a 1.5B-parameter model originally RL fine-tuned (from DeepSeek-R1-Distill-Qwen-1.5B (DeepSeek-AI et al., 2025)) on this dataset with a 24K token context length. Due to compute constraints, we restrict the maximum context length to 4K tokens during training and to 8K tokens during evaluation. The model is further fine-tuned for 700 steps with LCPO-Exact objective (Equation 1), and the resulting model is referred to as L1-Exact. The model is further RL finetuned for 120 steps with the objective mentioned in Equation 2, and the resulting model is referred to as L1-Max.

**Baselines.** We evaluate our proposed method against the following baselines:

**1.) DeepSeek-R1-Distill-Qwen-1.5B:** is the version of Qwen-2.5-1.5B-Instruct finetuned on reasoning traces of DeepSeek's R1 model. For brevity, we refer it as DeepSeek-R1-1.5B.

**2.) DeepScaleR-1.5B-Preview:** the original model, evaluated without any length control modifications. For brevity, we refer this model as DeepScaleR-24K.

**3.) DeepScaleR-1.5B-Preview-4K:** a version of DeepScaleR-24K fine-tuned with 4K context length. This is done due to computational constraints of training `LCPO` with long sequence length (such as 24K used in DeepScaleR-24K). The model therefore serves as a fair comparison to `L1`. For brevity, we refer to this model as DeepScaleR-4K.

**4.) S1:** (Muennighoff et al., 2025) is a budget-forcing method, which controls reasoning length using simple test-time interventions. Concretely, once the maximum token budget is reached, S1 stops further generation and instead inserts "Final Answer" in the prompt to force model to generate the final answer.

**Evaluation Protocol.** We evaluate our approaches along two dimensions. First, we assess the model's ability to adhere to the targeted length by reporting the mean deviation between the generated token length $n_y$ and the target $n_{gold}$. Second, we evaluate the overall performance (i.e., problem-solving accuracy) when generating responses at different target lengths. In our experiments, target lengths are selected from $\{512, 1024, 2048, 3600\}$ tokens. For all our experiments, we run 16 random seeds, with a temperature of 0.6.

**Hyperparameters and Implementation Details.** For GRPO training, we adopt the same hyperparameters as in DeepScaleR-1.5B-Preview. In particular, we use a learning rate of 1e-6 and a batch size of 128. The maximum context length is set to 4K tokens at training time and extended to 8K tokens during evaluation. Training is performed for 700 steps using the VeRL framework (MLSys, 2025). During training, the target length $n_{gold}$ is sampled uniformly from $U(n_{min}, n_{max})$, where we set $n_{min} = 100$ and $n_{max} = 4000$. The balancing parameter $\alpha$ in Equation 1 is fixed at 0.0003. Note that we did not conduct extensive hyperparameter tuning, so one can expect further improvements with additional optimization.

## 5 Results and Analysis

In this section, we report and analyze the effectiveness of the proposed method (`LCPO`) across various settings and benchmarks. We evaluate our method's relative performance, generalization capability on out-of-domain tasks, controllability of length constraints and competitive performance in short CoT setups, and examine learned reasoning patterns.

**L1 significantly outperforms other length-controlled models while maintaining strong performance.** Figure 2 compares performance of `L1-Exact` and `L1-Max` with other baselines across varying generation lengths. Both variants of `L1` achieve superior performance across all token budgets while maintaining strong length control. Compared to S1, the only other method specifically designed for length control, `L1` shows remarkable improvements, over 100-150% relative and 20-25% absolute performance gains at both 512 and 1024 token budgets. This substantial difference can be attributed to two key factors: (1) `L1` intelligently adapts its chain-of-thought to fit within specified length constraints without disrupting the reasoning process, while S1 often truncates mid-reasoning; and (2) `L1` is explicitly trained to generate high-quality reasoning chains of varying lengths, effectively distilling reasoning patterns from longer chains to shorter ones.

Moreover, with `L1`, we observe a log-linear scaling pattern, similar to the prior works O1 and S1 by OpenAI—performance improves linearly with respect to the log-length of generated reasoning chains. However, this scaling curve for `L1` exhibits a notably smaller slope (0.24 vs. 0.37 slope of S1), indicating substantially improved effectiveness at lower token ranges.

`L1-Exact` performs approximately 1% below DeepScaleR-4K, which is the same underlying model as `L1`, but trained without length constraints. However, this difference is primar-

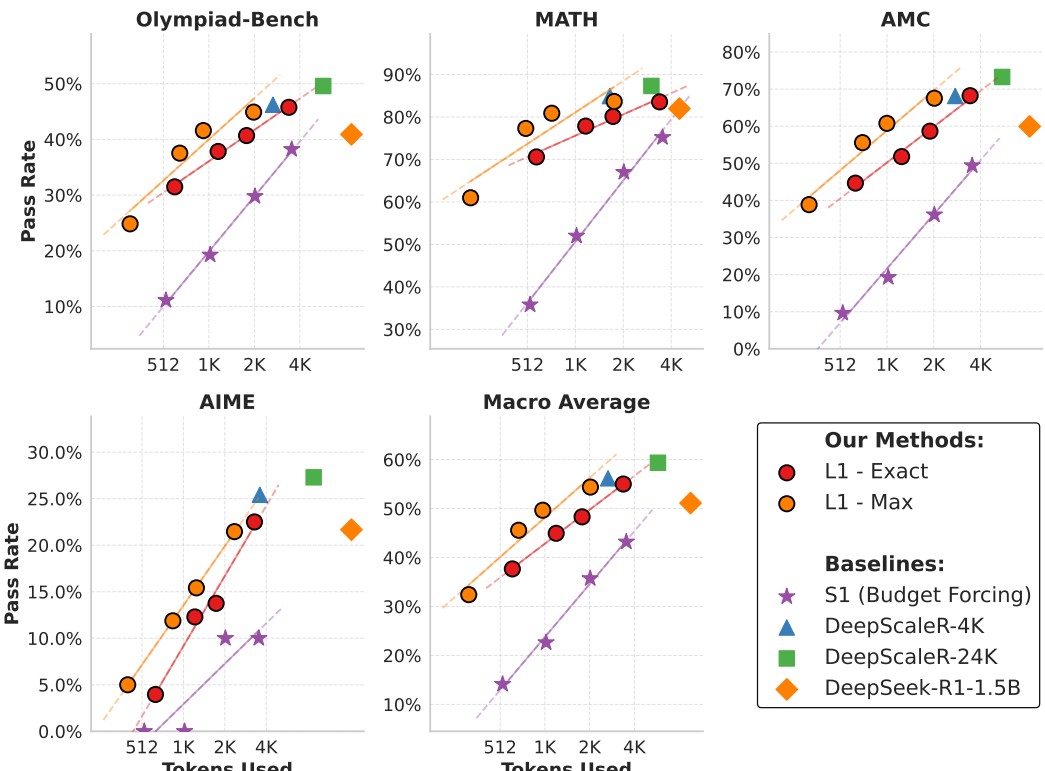

Figure 2: **Performance of L1 under varying token budgets.** Pass rate (y-axis) across benchmarks as a function of tokens used (x-axis). L1-Exact and L1-Max significantly outperform length-controlled baseline (S1) while approaching performance of unconstrained models.

ily observed in the AIME dataset, where unconstrained models can generate very long chains for complex problems. Additionally, L1-Exact allocates the same token budget to all problems regardless of difficulty, potentially using extra tokens on simpler problems. L1-Max effectively alleviates this challenge, matching the performance of DeepScaleR-4K by optimizing token usage based on problem difficulty while respecting the upper ceiling. In doing so, it outperforms even L1-Exact often by up to 2x fewer tokens. L1-Max is particularly valuable when exact token counts are less desirable than a worst-case compute budget. Finally, the scaling trends suggest that with longer context training, L1 would match or even surpass DeepScaleR-24K's performance while maintaining a high degree of length control.

**L1 generalizes effectively to out-of-domain (OOD) tasks.** We evaluate L1's ability to generalize length control capabilities to domains outside its RL training distribution. We categorize out-of-domain (OOD) datasets: general reasoning datasets GPQA and LSAT that were not explicitly used in L1's training but is likely within DeepSeek-R1-1.5B's training domain and MMLU, which likely falls even outside DeepSeek-R1-1.5B's training distribution.

Figure 3 confirms that L1 generalizes robustly to new domains: performance consistently scales positively with token budget for OOD general reasoning datasets, approaching or matching DeepScaleR-4K benchmarks despite explicit length control constraints. For GPQA and LSAT, we observe the same linear performance scaling trend as in our primary datasets, with L1 matching DeepScaleR-4K's performance at comparable token budgets. This generalization is particularly impressive given that L1 was not explicitly trained on these tasks. For MMLU, we see a less pronounced linear scaling relationship ($R^2 = 0.66$), likely because these knowledge-focused questions benefit less from extended reasoning.

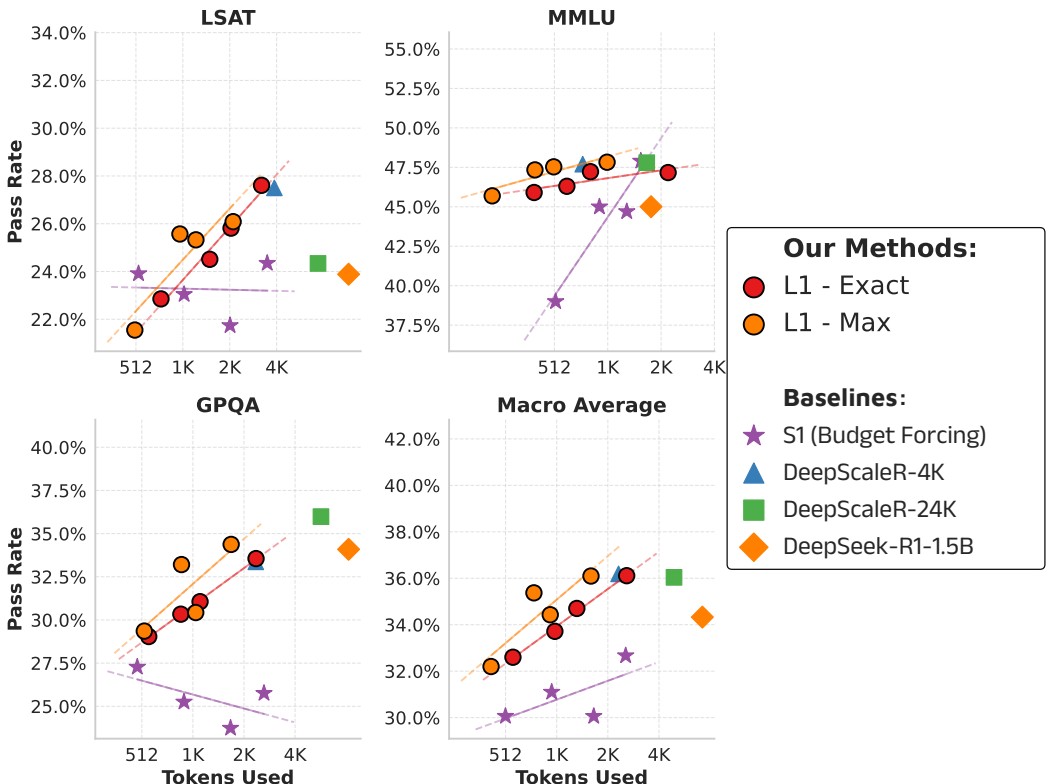

Figure 3: **Out-of-domain (OOD) generalization of L1.** L1 generalizes its length control to out-of-domain tasks, showing linear scaling of pass rate (y-axis) over generation length (x-axis) matching or exceeding performance of various baselines.

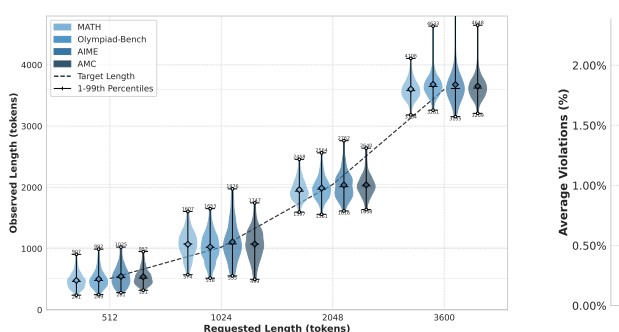

Figure 4: **Length control precision** across different datasets. The plot show distribution of response lengths at different requested lengths. Our model maintains consistent length control, with output lengths closely matching the requested lengths.

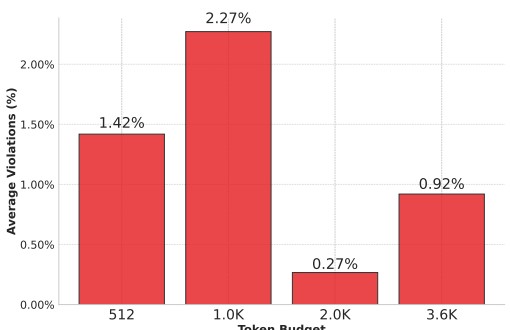

Figure 5: **Budget Violation Rates** across different datasets. L1-Max maintains less than 2.5% budget soft violation rate on across different token budgets, with lower violations at higher token lengths. On average the violation rate of mere 1.3%.

**L1 follows length constraints with high precision.** We quantitatively assess L1's ability to follow length constraints across various mathematical reasoning datasets. As shown in Figure 4, our model maintains consistent control across all token budgets (512, 1024, 2048, and 3600 tokens), with observed output lengths usually closely matching the requested lengths. Further, in Figure 11, we show the mean error: $(\frac{E_{x \sim D}[n_{generated}] - n_{gold}}{n_{gold}})$, which captures the

| Model | AIME | | MATH | | AMC | | Olympid-Bench | | Average | |
|---|---|---|---|---|---|---|---|---|---|---|
| | # Tokens | Accuracy | # Tokens | Accuracy | # Tokens | Accuracy | # Tokens | Accuracy | # Tokens | Accuracy |
| L1-Max | 401 | 5.0 | 299 | 70.8 | 431 | 47.1 | 410 | 33.4 | 385 | 39.1 |
| L1-Short | 422 | 12.1 | 321 | 75.4 | 395 | 46.7 | 390 | 36.0 | 382 | 42.6 |
| Qwen-1.5 | 877 | 7.7 | 554 | 73.4 | 791 | 47.5 | 784 | 35.4 | 752 | 41.0 |
| L1-Exact | 880 | 9.2 | 583 | 70.6 | 801 | 49.7 | 770 | **35.5** | 758 | 41.2 |
| L1-Max | 858 | **11.9** | 541 | **78.3** | 757 | **55.9** | 722 | **38.5** | 720 | **46.2** |
| Llama-3.3-70B | 1024 | 3.8 | 573 | 74.9 | 838 | 48.6 | 859 | 37.3 | 824 | 41.2 |
| L1-Exact | 983 | 10.6 | 583 | 70.6 | 801 | 49.7 | 913 | 36.9 | 820 | 41.9 |
| L1-Max | 1059 | **15.0** | 631 | **79.4** | 921 | **59.2** | 853 | **40.5** | 866 | **48.5** |
| GPT-4o | 867 | 10.0 | 591 | 78.4 | 990 | 51.8 | 911 | **42.4** | 840 | 45.6 |
| L1-Exact | 880 | 10.0 | 583 | 70.6 | 1060 | 51.3 | 913 | 36.9 | 859 | 42.2 |
| L1-Max | 858 | **11.9** | 631 | **79.4** | 921 | **59.2** | 853 | 40.5 | 816 | **47.8** |

Table 1: **Short CoT performance across math benchmarks.** L1-Max surpasses its non-reasoning base model and competes with frontier models under identical short CoT lengths.

average deviation from target lengths across the dataset. The figure demonstrates that mean error is low: close to 3% for all math reasoning datasets. Although OOD datasets exhibit predictably higher errors (20-40%), these remain preferable over uncontrolled prompting. Further Analysis in Appendix A.2 demonstrates that larger errors primarily appear at higher token budgets on tasks like MMLU, where the longer chain of thoughts is mostly unnecessary. Additionally, in Appendix A.1, we show that error can be further reduced significantly with extended RL training.

Further, for L1-Max, we show the budget violation rates in Figure 5. In particular, we compute soft violation rates: $|n_{generated} - n_{gold}| > 500$. The violation rates are low, ranging from 0.3% to 2.3% across different token budgets. The results demonstrate that L1-Max can be reliably used for ensuring length control.

**Long CoT Models are secretly Strong Short CoT Models.** Given L1's strong performance at lower token budgets, we conducted a focused evaluation comparing it to both its base non-reasoning model (Qwen-2.5-1.5B-Instruct) and significantly larger non-reasoning models (GPT-4o and Llama-3.3-70B) at comparable *generation lengths*. Table 1 presents these results, showing that L1 consistently outperforms or matches all models across all datasets despite using equivalent token budgets. Further, on average, L1 is 5% better than its non-reasoning counterpart, and even outperforms GPT-4o by 2% on average. We refer to these models as Short Reasoning Models (SRMs), given their shorter generation lengths, yet exhibiting similar patterns as full reasoning models (See Section 5).

This finding is remarkable, as to the best of our knowledge, this is the first demonstration that a 1.5B model can outperform frontier models such as GPT-4o, despite using the *same generation length*. Overall, the results signify that with suitable RL training, long CoT models can be adaptively used as short CoT models, while significantly outperforming their base counterparts at the same generation length.

**L1 employs distinct reasoning strategies at different token budgets.** To understand how L1 changes its reasoning approach across different length constraints, we analyzed how frequently certain reasoning-related terms appear in outputs of different lengths. Specifically, we calculated the normalized occurrence rate of most common reasoning terms in 512-token outputs compared to 4096-token outputs, showing how the model's reasoning strategies shift when given different length constraints. Figure 6 organizes these keywords into 4 distinct reasoning patterns: "Self-Correction and Verification," "Exploration and Alternatives," "Context Setting," and "Conclusion Drawing."

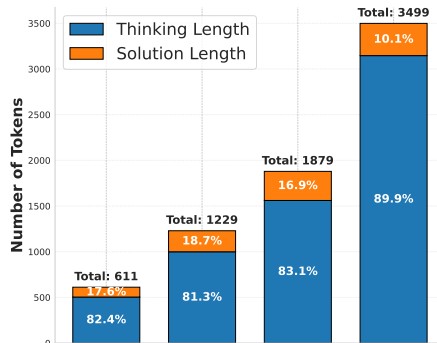

Figure 7: **Distribution of tokens for thinking vs. solution across chain-of-thought lengths.**

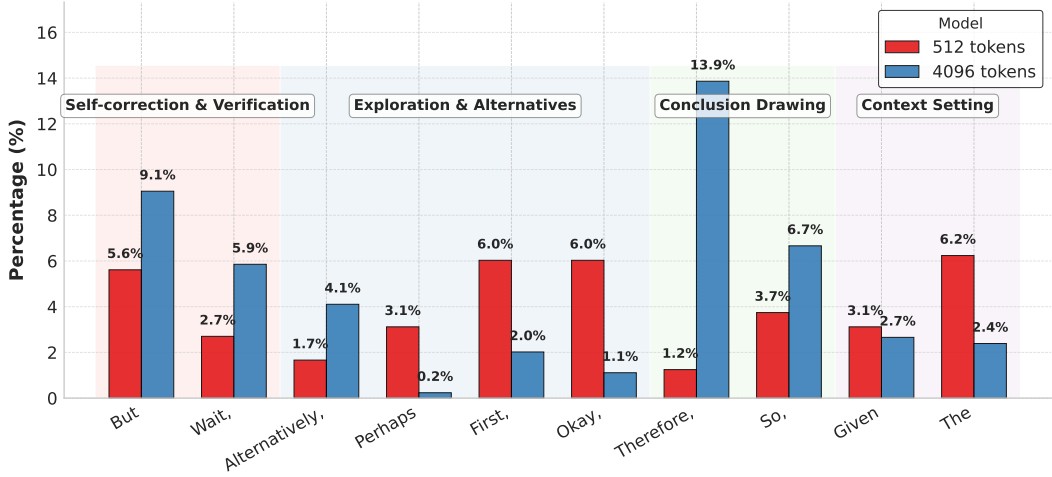

Figure 6: **Keyword usage by reasoning pattern at 512 vs. 4096 tokens.** L1 adaptively adjusts reasoning patterns based on requested token length.

Figure 6 shows that self-correction and verification keywords appear approximately twice as frequently in 4096-token outputs compared to 512-token outputs. Similarly, conclusion-drawing terms increase 2-10x with increased token budget, indicating more thorough solution validation and completion. Interestingly, most exploration-related keywords decrease in relative frequency at higher token counts, with "Alternatively" being a notable exception. Overall, we observe that smaller CoTs have reasoning patterns similar to their longer counterparts, but with changed relative frequencies that favor more self-verification and conclusion drawing in longer CoTs.

Further, Figure 7 shows the ratio of thinking tokens (those within <think> tags) to solution tokens for different generation lengths. We observe that the ratio is relatively stable across different generation lengths. This implies for shorter CoTs, the model generally provides short solutions (often just outputting the final answer), which helps save tokens. As generation length increases, we notice a stabilized response length in the last two bars, implying the model scales its thinking tokens without making the final solution overly verbose.

**Additional Experiments and Analysis** We further conduct several additional experiments and analysis to validate the robustness and generalizability of our approach across different configurations and model scales.

**Reward Function Ablations.** We systematically evaluate alternative reward function formulations for L1-Max in Appendix A.7 to understand the impact of different objective designs on both performance and length adherence. Our ablation study examines three variants: a single-objective approach using only the maximum length constraint, an additive formulation combining correctness and length penalties, and a sigmoid-based penalty function. While all variants achieve similar budget violation rates, the additive formulation exhibits undesirable behavior by collapsing to extremely short chain-of-thought sequences that trivially satisfy constraints but severely degrade reasoning quality. Our chosen multiplicative formulation in Equation 2 strikes an optimal balance, maintaining strong performance while ensuring robust adherence to length constraints.

**Sampling Parameter Robustness.** Our analysis in Appendix A.4 demonstrates that L1's length control capabilities remain stable across varying decoding parameters. When evaluated across temperature settings from 0.0 to 1.0, the model maintains consistent performance with mean length deviations of only 5.6-6.3% from target constraints. This robustness is particularly important for practical deployment, as it ensures reliable length control regardless of the specific sampling strategy employed. The observed average token lengths (1765-1782) closely match the theoretical expectation of 1796 tokens when averaged across our standard

evaluation lengths, indicating that the model's learned length control generalize effectively beyond the sampling parameters used in RL training.

**Parallel versus Sequential Scaling Trade-offs.** In Appendix A.8, we investigate how L1 performs when computational budget is allocated through parallel sampling (majority voting) versus sequential scaling (longer chain-of-thought generation). Our findings reveal that sequential scaling consistently outperforms parallel approaches at equivalent token budgets, aligning with recent trends in test-time compute allocation. Nonetheless parallel scaling is still beneficial, as it may be desirable in cases where lower latency is desired.

**Supervised Fine-tuning is not effective.** Our analysis in Appendix A.6 highlight ineffectiveness of supervised fine-tuning for inducing length control in reasoning models. In particular, we generate multiple responses from the reasoning model, relabel reasoning chains with their token lengths and repurpose this generated data for training. Our results highlight that SFT-trained models consistently fail to follow length constraints, generating very long output regardless of requested lengths. We hypothesize this failure stems from two factors: the narrow distribution of output lengths for given questions leads models to ignore length instructions, and the absence of online length-sensitive rewards prevents acquisition of adaptive reasoning capabilities. The result highlights the effectiveness of reinforcement learning based LCPO for achieving high degree of length control.

**Scaling to Larger Models.** We demonstrate the scalability of our approach by applying LCPO to DeepSeek-R1-Distill-7B in Appendix A.9, showcasing the method's potential for larger reasoning models. The 7B model exhibits identical trends to our 1.5B experiments: high length controllability, minimal budget violation rates, and strong performance relative to length-controlled baselines across token budgets.

# 6  Conclusion

In this work, we introduced Length Controlled Policy Optimization (LCPO), a simple yet powerful reinforcement learning-based method enabling adaptive control over the length of reasoning chains in language models. We use LCPO to train L1, a reasoning language model, optimized to produce outputs that adhere to length constraints given in its prompt. LCPO significantly surpasses previous test-time scaling methods, achieving over 100% relative and 20% absolute improvements in mathematical reasoning tasks compared to prior length-control approaches. Further, we demonstrated that L1 generalizes robustly beyond its training distribution, extending its length-controlled capabilities to out-of-domain tasks. Furthermore, our analysis revealed an intriguing phenomenon: models trained to generate longer reasoning chains become unexpectedly strong short reasoning models, outperforming significantly larger frontier models such as GPT-4o at identical generation lengths. By providing length control using simple prompt, LCPO opens promising avenues toward more efficient, flexible, and scalable reasoning models.

# 7  Limitations

Our results highlight strong length control capabilities of LCPO for both in-domain and out-of-domain tasks. However, we note that the models may not generalize to requested lengths that are longer than what they are trained for. It remains an important future direction to train models to use more inference compute at test than at training time. Further, we trained our models with length rewards on the complete outputs reflecting the real-world cost incurred by end-users. However, exploring other variants of rewards that, for instance, control the length of the reasoning tokens could be an interesting future direction.

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

# A Results

## A.1 Extended training further improves length constraint precision.

Table 2: Length Error Comparison Between Methods

| Method | Mean Error (%)↓ | RMSE Error (%)↓ |
|---|---|---|
| **Math Reasoning** | | |
| L1-Exact | **3.01** | 18.44 |
| L1-Exact + | 3.15 | **10.04** |
| **OOD-1 (General Reasoning)** | | |
| L1-Exact | 21.22 | 31.37 |
| L1-Exact + | **10.89** | **16.77** |
| **OOD-2 (General knowledge)** | | |
| L1-Exact | 40.54 | 42.61 |
| L1-Exact + | **23.57** | **33.48** |

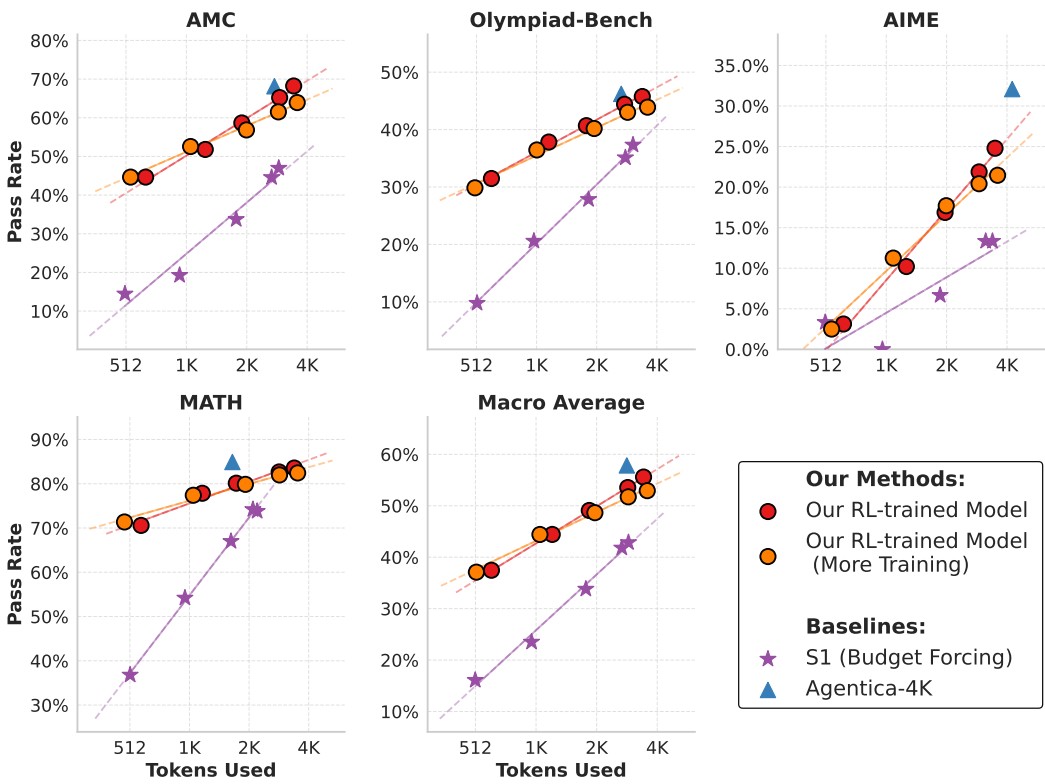

Figure 8: Performance vs Length for `LCPO` with more training

Motivated by the remaining gap in length control precision, we investigate if further RL training would improve length adherence. In particular, we further fine-tune L1-Exact for 500 RL steps. We refer to the derived model as L1-Exact+ Interestingly, Table 2 reveals substantial precision improvements—particularly steep RMSE reductions—from additional training across both math and OOD datasets. Yet, greater length precision appears to slightly lower performance at high token ranges (Figure 8). This effect likely stems from the model's stricter adherence to length constraints, which reduces its flexibility to generate longer CoTs to more challenging problems. Nevertheless, the model maintains its overall performance trends, suggesting an interesting direction for future work: enabling users to dynamically

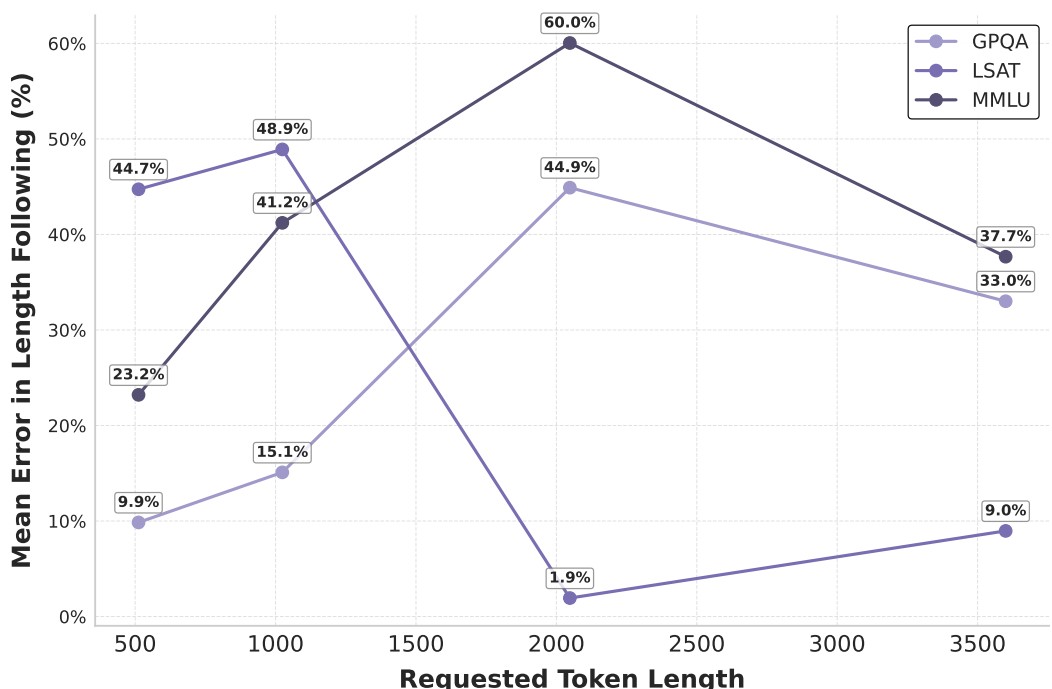

Figure 9: Length error vs token length for OOD datasets. Datasets like MMLU show higher error for longer chains, likely because longer chains are not particularly useful for these datasets.

control the tradeoff between strict length adherence and optimal performance based on their specific requirements.

## A.2 `LCPO` follows length constraints with high precision.

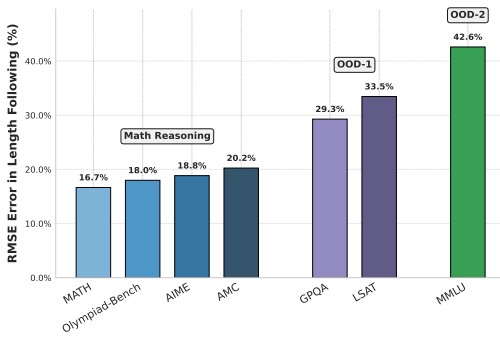

Figure 10: **RMSE in Length of generated chains** across different datasets. `LCPO` exhibits low RMSE for math reasoning datasets, maintaining high precision in length control. OOD datasets exhibit higher RMSE, particularly MMLU, where longer chains are less necessary.

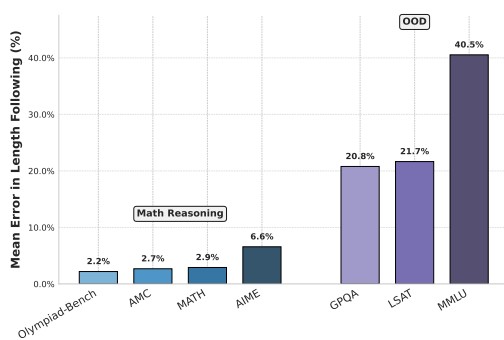

Figure 11: **Mean Error in Length of generated chains** across different datasets. `L1` maintains around 3% average deviation on math reasoning datasets and higher but acceptable deviation on OOD tasks.

In Section 5, we show that `LCPO` follows length constraints with high precision, although with larger errors in OOD datasets. In this section, we particularly analyze the generation

length at which error occurs. Figure 9 shows error as a function of generated token length for 3 different OOD datasets.

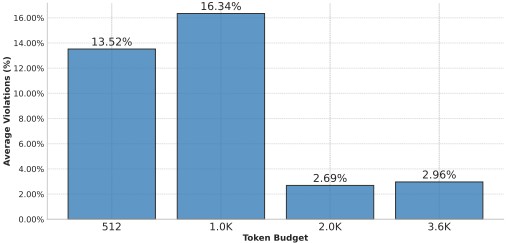

Figure 12: **Hard Budget Violation Rates** across different datasets. L1 maintains relatively higher, but acceptable violation rates of 9% on average across different token budgets.

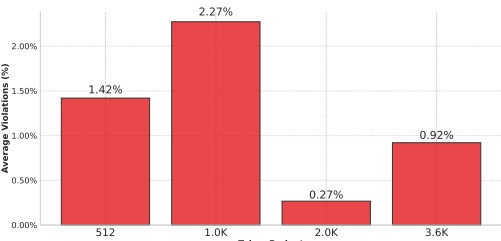

Figure 13: **Soft Budget Violation Rates** across different datasets. L1 maintains less than 3% budget violation rate on across different token budgets.

Further, we also report both soft- and hard-violation rates in Figure 12 and Figure 13. Notably, while soft-violation rates are very low ($\leq$ 3%), hard-violation rates are relatively higher (9% on average across different token budgets). The violation rates are particularly higher at lower token budgets, as the model prioritizes performance over length control, and generating correct solutions with shorter CoTs is relatively more difficult.

| Model | AIME2024 | AIME2025 | Delta |
|---|---|---|---|
| Qwen-Math-1.5 | 12.9 | 7.7 | -5.2 |
| GPT-4o | 6.7 | 10.0 | +3.3 |
| LLama-3.3-70B | 27.3 | 3.8 | -23.5 |
| DeepSeek-R1-1.5B | 29.2 | 21.7 | -7.5 |
| DeepScaleR-24K | **40.2** | 27.3 | -12.9 |
| DeepScaleR-4K | 32.1 | 25.4 | -6.7 |
| L1-Exact *(ours)* | 24.8 | 22.5 | -2.3 |
| L1-Max *(ours)* | 27.3 | 21.5 | -5.8 |
| L1-Exact *(1024 tokens)* | 10.2 | 12.3 | +2.1 |
| L1-Max *(1024 tokens)* | 16.3 | 11.9 | -4.4 |

Table 3: Comparison of model performance on AIME2024 and AIME2025 benchmarks.

**Memorization or Generalization?**   Given the surprisingly strong performance of LCPO, particularly at lower token budgets, a natural question arises regarding whether the model has memorized solutions, suggesting potential train-test leakage. To address this concern, we specifically evaluate on the AIME2025 dataset, an exam released after the training of all evaluated models. Results in Table 3 demonstrate that the performance drop for LCPO is substantially smaller ($\approx$ 2%) compared to other models such as DeepScaleR-24K (-12.9%), DeepSeek-R1-1.5B (-7.5%), and even Llama-3.3-70B (-23.5%). These results provide strong evidence that despite using shorter CoT lengths, our model demonstrates generalization rather than relying solely on memorization.

### A.3   Aha Moment in L1's training

Figure 14 shows the training logs of L1-Exact. During the first 300 RL steps, we observe that the model prioritizes improving its average solve rate while the minimum response length continues to decrease. This suggests that the model initially focuses on correctly generating solutions rather than strictly adhering to length constraints. However, at approximately 300 RL steps, we observe a significant phase transition in the training dynamics. Specifically, the reward and token adherence score (reward − solve_mean) begin to increase rapidly, while

| Temperature | Mean Deviation | Solve Rate (%) | Avg. Token Length |
|---|---|---|---|
| 0.0 | 6.0% | **46.6%** | 1778 |
| 0.3 | 6.3% | **46.6%** | **1782** |
| 0.6 | 6.2% | 46.5% | 1778 |
| 1.0 | **5.6%** | 46.4% | 1765 |

Table 4: Effect of sampling temperature on length control and accuracy for L1-Exact. Despite changing the temperature, the model shows similar errors in length control and similar Pass@1 solve rates throughout all temperature settings.

the minimum response length exhibits a sharp drop, eventually converging at around 100 tokens (the lower bound of $n_{gold}$). This distinct transition point represents when the model effectively learns the concept of token adherence and subsequently continues to refine this capability throughout further training.

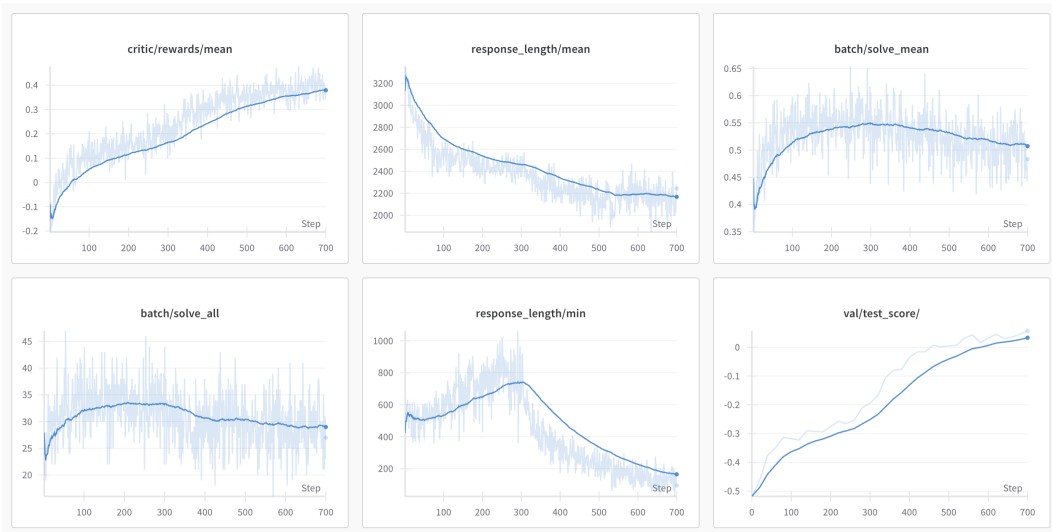

Figure 14: Training Logs for LCPO-Exact. We notice an Aha Moment, where the model starts adhering to token constraint after 300 RL steps, as visible in peaking solve rate, and sharp drop in minimum token length.

## A.4 Does L1 generalize to varying sampling parameters

We evaluate L1-Exact's robustness to different sampling parameters by varying the temperature during inference. Table 4 presents results across temperature values ranging from 0.0 to 1.0. We observe that the model maintains consistent performance in terms of both solve rate and length control accuracy. Specifically, the mean deviation from target length remains stable (5.6-6.3%) across all temperature settings, with marginal differences in token length and solve rate. These results indicate that L1's length control capabilities are robust to sampling parameters, allowing flexible deployment across different inference settings without compromising performance or adherence to length constraints. Further, given that we average across four target lengths $n_{gold} \in \{512, 1024, 2048, 3600\}$, the observed average token lengths (1765-1782) closely match the ideal average of 1796 tokens.

## A.5 Prior Reasoning Models do not follow length constraints in Prompts

A natural question is whether existing reasoning models can follow length constraints through simple prompting without specialized training. To investigate this, we explicitly prompted DeepScaleR-24K to generate responses of specific token lengths (512, 1024, 2048,

| Requested Length | 512 | 1024 | 2048 | 3600 |
|---|---|---|---|---|
| Generated Length | 5491 | 6072 | 5846 | 6421 |

Table 5: Average length generated by DeepScaleR-24K for different requested lengths. The model does not follow the user-specified length constraint.

and 3600) using clear prompts like "Think for exactly N tokens." As shown in Table 5, the model consistently generates outputs close to 6000 tokens regardless of the requested length, demonstrating a complete insensitivity to length instructions.

### A.6 Supervised Fine-Tuning does not learn length control

We further compare how our proposed RL recipe compares to SFT. For training models with supervised finetune, we first generate responses without any length control, then measure their token lengths and re-prompt with the measured token counts to create instruction-response pairs with explicit length requirements. For training the models, we perform a hyperparameter search over Learning Rates: 1e-6, 1e-5, 1e-4 and Batch Sizes: 256, 512. We use validation loss for early stopping.

After training with this SFT data, we evaluated adherence by prompting for specific lengths. The results are presented in Table 6. The results highlights that the model still failed to follow length constraints, producing very long outputs regardless of the requested target.

| Requested Tokens | 512 | 1024 | 2048 | 4096 |
|---|---|---|---|---|
| Actual Tokens | 21388 | 22749 | 21426 | 20903 |

Table 6: SFT-only model does not learn to follow token-length constraints despite being trained on relabeled data with explicit length requirements.

These results, along with Table 5, indicate that offline methods such as SFT are ineffective at inducing precise length control in reasoning models. We hypothesize two reasons: (1) the narrow distribution of output lengths for a given question leads the model to ignore length instructions, and (2) without an online length-sensitive reward, the model does not acquire the capability to adjust its reasoning to meet target budgets.

### A.7 Ablating Reward Functions in `LCPO`

We evaluate how different objective functions affect the performance and token adherence of `LCPO`. Specifically, we ablate Equation 2 using the following variants:

1. **L1-Max Single Objective**: During the training of L1-Max, we originally employed both objectives from Eq. 2 and Eq. 1. In this variant, we exclusively use the former.
2. **L1-Max Addition**: This variant implements a simple addition of correctness and length penalty objectives instead of multiplication. The objective function is: $r(y, y_{gold}, n_{gold}) = \mathbb{I}(y = y_{gold}) - \alpha \cdot |n_{gold} - n_y|$.
3. **L1-Max Sigmoid**: This variant applies a sigmoid function to penalize length violations using the objective: $r(y, y_{gold}, n_{gold}) = \mathbb{I}(y = y_{gold}) \cdot \sigma(\alpha \cdot (n_{gold} - n_y))$, where $\sigma(x) = \frac{1}{1+e^{-x}}$ is the sigmoid function.

We train each variant for 120 RL steps, initializing from the same L1-Exact checkpoint. Figure 17 presents the budget violation rates across these variants. While all variants exhibit similar violation rates across different token budgets, L1-Max Addition achieves a 0% budget violation rate consistently. This perfect adherence, however, occurs because the model collapses to generating extremely short CoTs, which trivially satisfies the maximum length constraint which is undesirable behavior.

Figure 15 demonstrates this effect by showing performance versus token length for all variants. LCP0-Max Addition performs substantially worse than the other three variants. For clarity, we exclude this variant in Figure 16. The remaining three variants perform similarly without any statistically significant differences. We ultimately select the standard L1-Max for all our experiments due to its simplicity and the potential advantages of its dual objective approach.

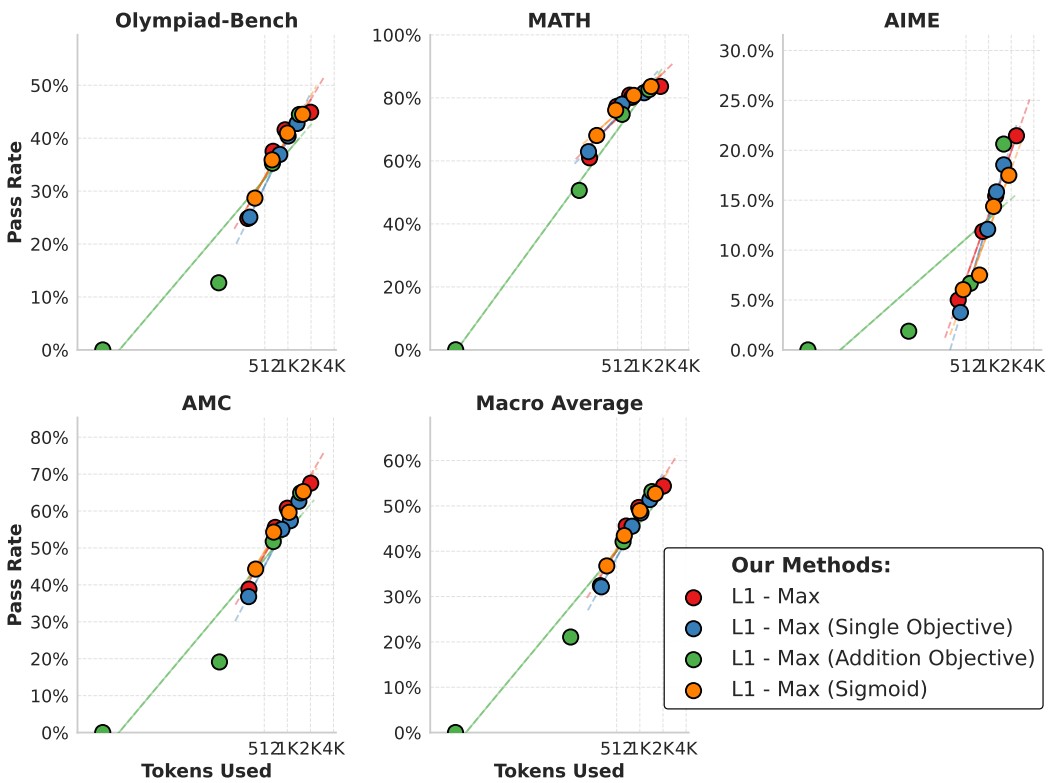

Figure 15: Performance of different Reward Functions (RFs) used to train L1-Max.

## A.8    Parallel vs Sequential Scaling

Figure 18 compares how sequential scaling (increasing CoT length) and parallel scaling (Majority Voting) vary in performance for the same incurred cost (total tokens generated). There are a few interesting observations from the graph: First, parallel scaling is almost always worse than sequential scaling, when compared at the same token budget. Second, parallel scaling can provide significant improvements, to all reasoning models (eg, more than 10% in some cases). Third, while Qwen-2.5-1.5B-Math, barely benefits from majority voting, L1-Max (1024 tokens), despite using similar number of generation tokens in each task, benefits significantly. This is noteworthy, since for many cases, it is desirable to have low token lengths, because of both reduced latency, and avoiding the quadratic cost of generation. In such cases, LCP0-Max clearly severs as a better alternative to parallel scaling of non-reasoning models. Finally, LCP0-Max (3600 tokens) matches the performance of DeepScaleR-4K baseline across parallel scaling. We do note a small dip in the last point, but this is because only 1 seed was used for parallel scaling, and as such the difference is not statistically significant.

## A.9    LCP0 scales to larger reasoning models

We further scaled the LCP0 recipe to DeepSeek-R1-Distill-7B model. Results are presented in Figure 19. We observe the same trends as with the 1.5B model: high controllability, low

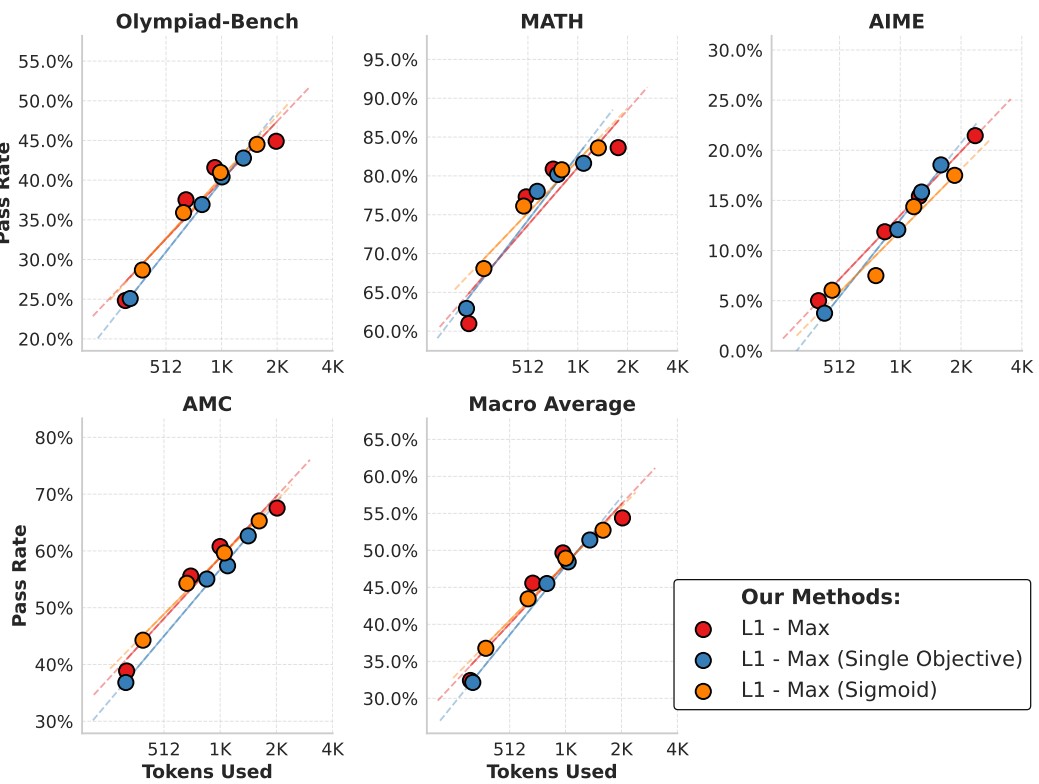

Figure 16: Performance of different Reward Functions (RFs) used to train L1-Max.

budget-violation rates, and strong performance relative to length-controlled baselines across budgets. These results highlight the effectiveness and potential of LCPO to scale to even larger reasoning models.

## A.10 Qualitative Examples

In Figures 20 and 21, we show two examples of L1-Exact solving the same problem but with different requested generation lengths: 512 and 3600 tokens. While the model with 512 tokens is unable to solve the problem correctly, the model with longer generation arrives at the correct answer. We refer interested readers to our website, which has more such interesting examples.

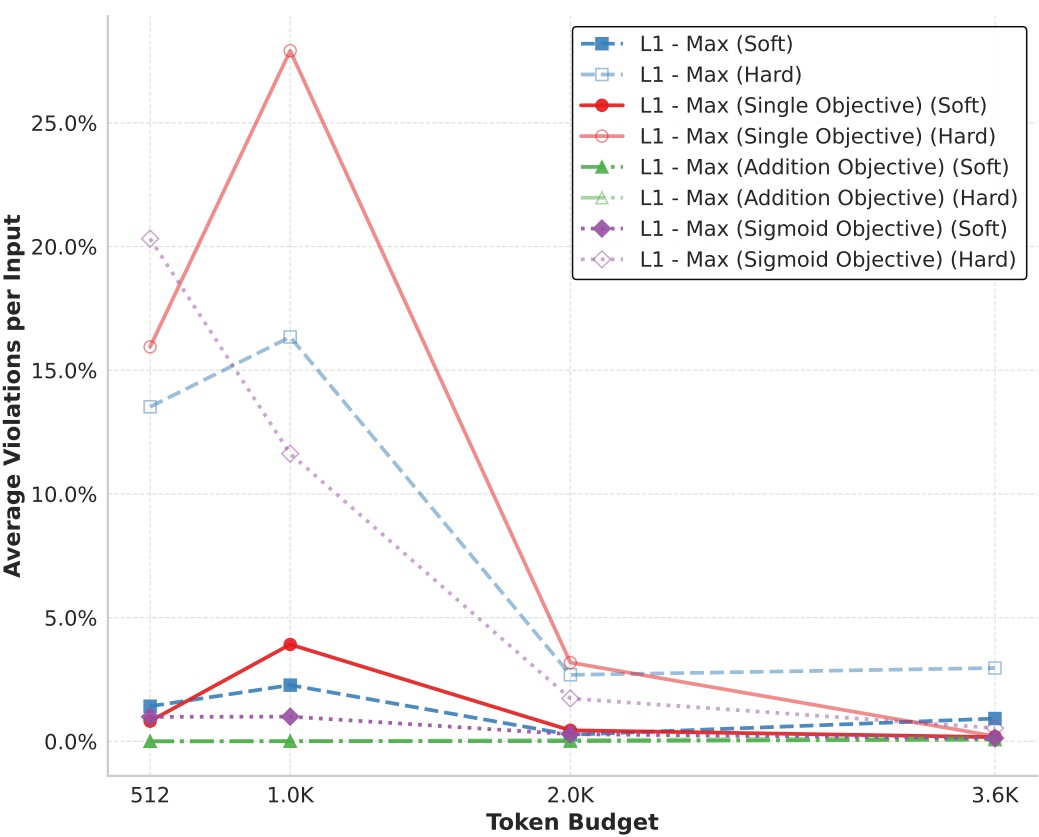

Figure 17: Budget Violation Rates for different Reward Functions used to train LCPO-Max.

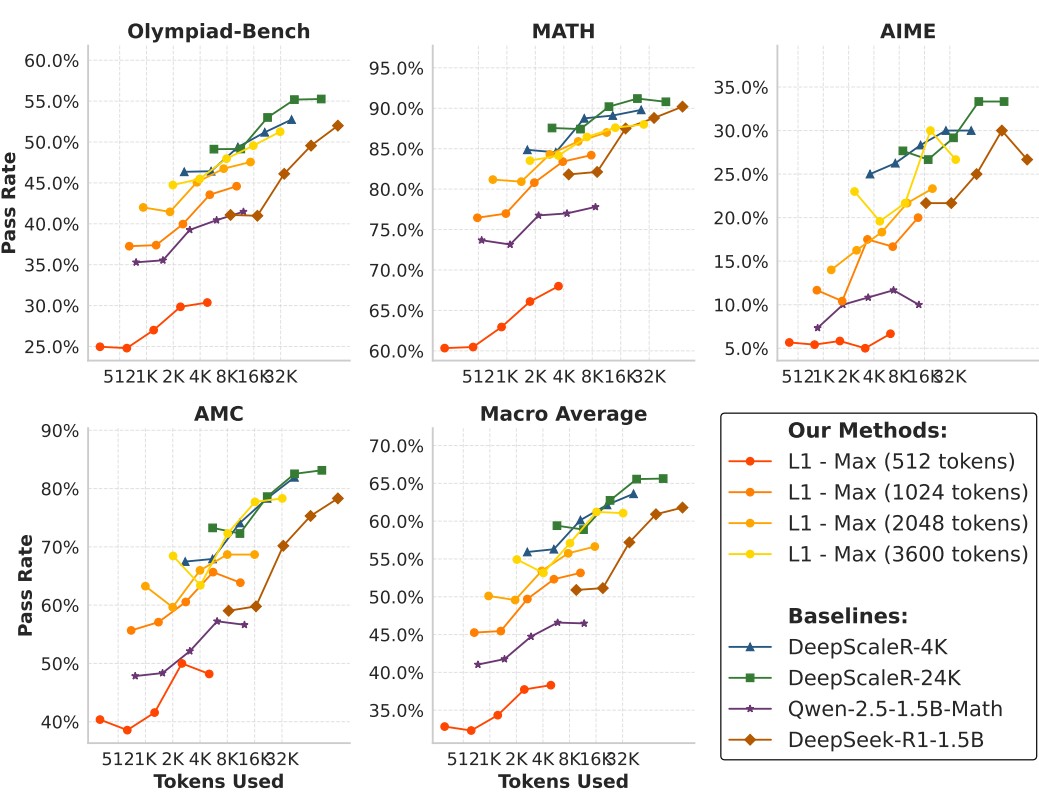

Figure 18: Parallel vs Sequential Scaling

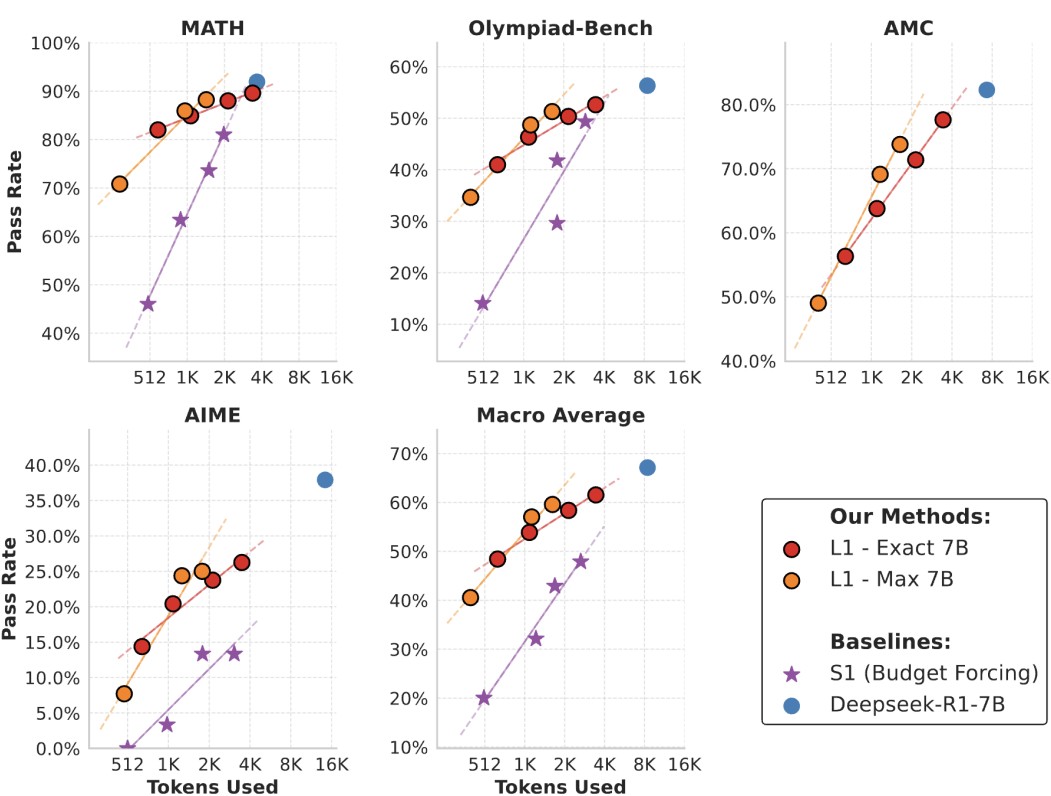

Figure 19: **Scaling `LCPO` to a 7B reasoning model.** We apply `LCPO` to DeepSeek-R1-Distill-7B reasoning model and show strong length control and performance on math reasoning tasks.

**Example**

**Question**

Find the largest possible real part of $[(75 + 117i)z + \frac{96+144i}{z}]$ where $z$ is a complex number with $|z| = 4$. Let's think step by step and output the final answer within boxed{}. Think for 512 tokens.

**Correct Answer**

$\boxed{540}$

**Correctness**

✖ **Incorrect**

**Tokens Requested**

512

**Model Response**

> **Thinking Process**
>
> Okay, so I have this problem: I need to find the largest possible real part of the expression $(75 + 117i)z + (96 + 144i)/z$, where $z$ is a complex number with $|z| = 4$. Hmm, let's break this down.
> First, since $|z| = 4$, $z$ can be written as $4e^{i\theta}$ for some angle $\theta$. That might help simplify things.
> Let me denote $z = 4e^{i\theta}$. Then, $1/z = (1/4)e^{-i\theta}$ because $|z| = 4$.
> So, the expression becomes $(75 + 117i) \cdot 4e^{i\theta} + (96 + 144i) \cdot (1/4)e^{-i\theta}$.
> Simplifying, that's $4(75 + 117i)e^{i\theta} + (24 + 36i)e^{-i\theta}$.
> Now, I need to find the real part of this expression. Let's denote $\text{Re}[(75 + 117i)z + (96 + 144i)/z]$.
> Breaking it down, $\text{Re}[(75 + 117i)z] + \text{Re}[(96 + 144i)/z]$.
> Since $z = 4e^{i\theta}$, $\text{Re}[(75 + 117i) \cdot 4e^{i\theta}] + \text{Re}[(96 + 144i) \cdot (1/4)e^{-i\theta}]$.
> This becomes $4\text{Re}[(75 + 117i)e^{i\theta}] + (1/4)\text{Re}[(96 + 144i)e^{-i\theta}]$.
> Now, let's compute $\text{Re}[(75 + 117i)e^{i\theta}]$. This is $75\cos\theta + 117\sin\theta$.
> Similarly, $\text{Re}[(96 + 144i)e^{-i\theta}]$ is $96\cos\theta - 144\sin\theta$.
> So, putting it all together, the real part is $4(75\cos\theta + 117\sin\theta) + (1/4)(96\cos\theta - 144\sin\theta)$.
> Simplifying, that's $300\cos\theta + 468\sin\theta + 24\cos\theta - 36\sin\theta$.
> Combining like terms, we get $(300 + 24)\cos\theta + (468 - 36)\sin\theta = 324\cos\theta + 432\sin\theta$.
> Now, we need to find the maximum of $324\cos\theta + 432\sin\theta$. This can be written as $R\cos(\theta - \phi)$, where $R = \sqrt{324^2 + 432^2}$.
> Calculating $R$: $324^2 = 104,976$, $432^2 = 186,624$. Sum is $391,600$. $R = \sqrt{391,600} \approx 625.6$.
> Thus, the maximum real part is approximately 625.6.

The largest possible real part is $\boxed{625.6}$.

Figure 20: Example of model response with 512 tokens. Note that the model's answer is incorrect.

---

**Example**

**Question**

Find the largest possible real part of $[(75+117i)z + \frac{96+144i}{z}]$ where $z$ is a complex number with $|z| = 4$. Let's think step by step and output the final answer within $boxed\{\}$. Think for 3600 tokens.

**Correct Answer**

$\boxed{540}$

**Correctness**

✔ **Correct**

**Tokens Requested**

3600

**Model Response**

> **Thinking Process (not shown to user)**
>
> Okay, so I have this problem: I need to find the largest possible real part of the expression (75 + 117i)z + (96 + 144i)/z, where z is a complex number with $|z| = 4$. Hmm, that seems a bit involved, but let's break it down step by step. First, let me understand what is given. We have a complex number z with magnitude 4, so z can be written in the form z = $4e^{i\theta}$, where $\theta$ is the argument of z. That might be useful because it allows us to express z in terms of its angle, which could simplify the expression.
>
> The expression we need to consider is $(75 + 117i)z + \frac{96+144i}{z}$. We are asked to find the largest possible real part of this expression. So, let's denote the expression as $E = (75 + 117i)z + \frac{96+144i}{z}$.
>
> [Reasoning process truncated for brevity ...]
>
> Then, adding $104,976 + 186,624 = 291,600$. $\sqrt{291,600} = \sqrt{2916 * 100} = \sqrt{2916} * 10$. Now, $\sqrt{2916}$: let's see, $54^2 = 2916$, because $50^2 = 2500, 54^2 = 2916$. So, $\sqrt{2916} = 54$, thus $\sqrt{2916 * 100} = 54 * 10 = 540$. So, that seems correct. Therefore, the maximum value of $Re(E)$ is 540.
>
> Therefore, the largest possible real part is 540.

Given the expression $(75+117i)z + \frac{96+144i}{z}$ where $z$ is a complex number with $|z| = 4$, we need to find the largest possible real part of this expression.

First, we express $z$ as $4e^{i\theta}$ since $|z| = 4$. Then, we substitute $z$ and $\frac{1}{z}$ into the expression:

$$(75 + 117i)z + \frac{96 + 144i}{z}$$

[Output truncated for brevity ...]

Thus, the largest possible real part is:

$$\boxed{540}$$

---

Figure 21: Example of model response with 3600 tokens. With more tokens, the model correctly calculates the answer.

