# OpenReview forum: "L1: Controlling How Long A Reasoning Model Thinks With Reinforcement Learning"
_colmweb.org/COLM/2025/Conference — COLM 2025_

### Official Review · Reviewer_E128 · 2025-04-16

**Rating:** 7
**Confidence:** 4
**Ethics Flag:** 1

**Summary:**

L1: Controlling How Long A Reasoning Model Thinks With Reinforcement Learning

Controlling the length of CoT reasoning chains is an important topic
LCPO optimizes for accuracy, and adheres to pre-specified length. 2 objectives? What is reward? Sum of two terms.
L1 is trained with LCPO.

Furthermore, out-of-distribution results.
Short-reasoning-models.

Muennighoff 2025 proposes the S1 method. LCPO improves on S1

**Questions To Authors:**

L1 is trained with LCPO.
However, CoT is test time. So how train with LCPO? Test/Train???
Please explain more clearly.

Provide explanation why OOD generalization occurs.
Violin plots in Figure 4 should be bigger. Very hard to read.

**Reasons To Accept:**

The topic is timely and important.

Experimental results look convincing: L1 shows significant improvement over S1 in Figure 2, explanation on page 6 is interesting.
OOD results (Figure 3) are nice.

Especially nice is the insightful explanation on page on why Long CoT Models are secretly Strong Short CoT Models (page 8).

Experiments use different versions of DeepSeek. Good.

**Reasons To Reject:**

Few reasons to reject, other than that one can always ask for more experiments.
What I really appreciate are that this paper tries to interpret the results, and gives nice intuitions  on page 6 and 8, 9 to explain their findings.
Only used a single LLM.

---

> ### Author Response · Authors · 2025-06-03
>
> We thank the reviewer for their thoughtful review and for appreciating the intuitions and interpretability efforts in our results. We address your questions below:
>
> ---
>
> >What I really appreciate are that this paper tries to interpret the results, and gives nice intuitions on page 6 and 8, 9 to explain their findings. Only used a single LLM.
>
> We are glad you liked the analysis. Based on your feedback, we have also trained a larger 7B model (Deepseek-r1-distilled-7B). The results can be found [here](https://anonymous.4open.science/r/L1-COLM-7832/results_7b/README.md).
>
> We see similar trends as 1.5B model: **High controllability**, **strong performance compared to baselines**. These results highlight the effectiveness and potential of LCPO to scale to larger model sizes.
> We will include more information about this in the main paper.
>
> ---
>
> > L1 is trained with LCPO. However, CoT is test time. So how train with LCPO? Test/Train??? Please explain more clearly.
>
> To clarify, we train L1 with prompts specifying various token lengths. At test time, length control is achieved simply by specifying the length in the prompt. For exact prompt template you can check Line 149 and qualitative examples in Figure 19, 20.
>
> ---
>
> > Provide explanation why OOD generalization occurs.
>
> We hypothesize that LCPO-trained models effectively learn the concept of length control independent of domain specificity. Recent studies (e.g., “SFT Memorizes, RL Generalizes” [1]) suggest RL with CoT training leads to better generalization in reasoning tasks compared to standard supervised finetuning, and further, our LCPO formulation that disentangles Length and Correctness rewards separately may have further led to strong generalization. That being said, providing a full explanation of why OOD generalization occurs would be interesting and valuable future work.
>
> ---
>
> > Violin plots in Figure 4 should be bigger. Very hard to read.
>
> Thanks for the feedback! We will update the figures in the camera-ready.
>
> ---
>
> Thank you again for your encouraging feedback and thoughtful suggestions; we are happy to clarify any further points you might have.

---

> > ### Comment · Reviewer_E128 · 2025-06-03
> > **Thank you**
> >
> > Thank you for the feedback.
> > You answered my questions and my score remains unchanged.

---

### Official Review · Reviewer_2aPy · 2025-05-10

**Rating:** 7
**Confidence:** 4
**Ethics Flag:** 1

**Summary:**

The paper studies the effectiveness of controlling the thinking length of reasoning language models (e.g., R1, o1) using reinforcement learning. Specifically, the paper introduces Length Controlled Policy Optimization (LCPO), a reinforcement learning approach that rewards a reasoning language model based on both the answer correctness and the output token length. In experiments, the paper considers DeepScaleR-1.5B-Preview as the base model and applies LCPO to produce two variants, L1-Exact for generating answers with the exact thinking length, and L1-Max for generating answers with a maximum thinking length. Both variants show a log-linear scaling pattern on both in-distribution and out-of-distribution tasks. They also outperform the s1 baseline.

**Questions To Authors:**

1. What are the values of $\alpha$ in Eqns 1 and 2?
2.  In Eqn 2, the range of 0-1 in the clip function makes me wonder if the amount of $n_{gold}-n_y$ is normalized in practice (e.g., $(n_{gold}-n_y) / n_{gold}$). Can the authors clarify the formulation further?
3. Why does s1 show a downward trend on GPQA and a flat curve on LSAT?
4. What is the experimental procedure of Table 1? Why are there three sets of L1 results on the same dataset? Is the "#Tokens" specified in the control instruction?

**Reasons To Accept:**

1. The paper explores an interesting and timely topic of controlling the chain-of-thought thinking length of reasoning language models.
2. The exploration of reinforcement learning algorithms is adequate, and the results (particularly the scaling patterns) are solid.
3. The paper includes further analyses, such as the precision of the length control and the budget violation rate of L1-Max, which are pretty interesting.

**Reasons To Reject:**

1. In LCPO, while the control instruction says “**Think for**_ $n_{gold,i}$ tokens”, the reward is calculated based on the **total output** length, rather than the thinking token length (i.e., text wrapped by <think>). The results in Fig 7 relieve my concern, but I still wonder why the reward is not based on comparing the exact thinking length with $n_{gold,i}$.
On the other way around, is the LCPO algorithm sensitive to the specific control instruction design?

2. Concern about the generalizability of LCPO's length control: The two L1 models were said to be trained under a 4k-token context and then evaluated under an 8k-token context. However, the results presented in Figs 2 and 3 do not include any data points where the L1 models used more than 4k tokens. It seems to imply that, because the L1 models were trained to generate at most 4k tokens, they cannot follow the test-time control for using more than 4k tokens.

3. A few language corrections:
- L199: "Agentic-24K" is not introduced
- Legend of Fig 3 needs update: "Agentica-4K", "Agentica-24K"

---

> ### Author Response · Authors · 2025-06-03
>
> We thank the reviewer for their thoughtful review! We appreciate that you find our work interesting and timely, our results solid, and the analysis interesting. We address your questions below:
>
> ---
>
> > In LCPO, while the control instruction says “Think for_
> ngold,i
>  tokens”, the reward is calculated based on the total output length, rather than the thinking token length (i.e., text wrapped by <think>). The results in Fig 7 relieve my concern, but I still wonder why the reward is not based on comparing the exact thinking length with
> ngold,i
> . On the other way around, is the LCPO algorithm sensitive to the specific control instruction design?
>
>
> We appreciate this insightful point. We chose total output length practically, as this often aligns better with real-world usage scenarios (the user owns the total cost and not just reasoning length). This choice performs well and is expected to work for different instruction designs (as evident from performance in L1-Exact and L1-Max settings). Nonetheless, exploring reward formulations explicitly targeting thinking tokens could indeed be an interesting variation.
>
> ---
>
> > Concern about the generalizability of LCPO's length control: The two L1 models were said to be trained under a 4k-token context and then evaluated under an 8k-token context. However, the results presented in Figs 2 and 3 do not include any data points where the L1 models used more than 4k tokens. It seems to imply that, because the L1 models were trained to generate at most 4k tokens, they cannot follow the test-time control for using more than 4k tokens.
>
> Indeed, our experiments confirm LCPO-trained models do not readily generalize to token lengths beyond those seen during training. Exploring strategies to improve length generalization is a promising direction for future research. We will add this discussion to the camera-ready.
>
> ---
>
> > A few language corrections:
>
> Thanks for pointing them out! We will fix them in the revision.
>
> ---
>
>
> > What are the values of
> α
>  in Eqns 1 and 2?
> In Eqn 2, the range of 0-1 in the clip function makes me wonder if the amount of
> ngold−ny
>  is normalized in practice (e.g.,
> (ngold−ny)/ngold
>
> We use α = 0.0003 in both equations. The term (ngold - ny) is not normalized; instead, we empirically chose α to balance correctness and length adherence without explicit normalization. We will clarify this further in the updated version.
>
> ---
>
> > Why does s1 show a downward trend on GPQA and a flat curve on LSAT?
>
> We note that both datasets are difficult for 1.5B model. Further, S1 often truncates the reasoning in the middle. Given the low performance (close to random guessing), we note that for S1 on both GPQA and LSAT difference in trends is not statistically significant.
>
> ---
>
> > What is the experimental procedure of Table 1? Why are there three sets of L1 results on the same dataset? Is the "#Tokens" specified in the control instruction?
>
> Our goal is to compare our models against state-of-the-art models at comparable token lengths. We evaluated multiple (3) baseline models, noted their token lengths, and for each length, we prompted the L1-Exact model to generate with the same token length. Therefore, L1 is listed three times. We will clarify this better in the revised paper.
>
> ---
>
> Thanks again for your thoughtful review and feedback. We look forward to addressing any other concerns you may have!

---

> > ### Comment · Reviewer_2aPy · 2025-06-05
> >
> > Thanks for the response! I don't have further concerns, but I suggest the authors discuss the token length generalization weakness carefully in the paper and share insights about how future work could tackle this issue.

---

### Official Review · Reviewer_ivGY · 2025-05-11

**Rating:** 7
**Confidence:** 4
**Ethics Flag:** 1

**Summary:**

The authors introduce Length Controlled Policy Optimization (LCPO), a rl technique to control the length of cot reasoning in language models. By training models (L1-Exact and L1-Max) with a reward function that balances task accuracy and adherence to a prompted length constraint, they achieve a controllable trade-off between computational cost (token generation) and reasoning performance. Key findings include outperforming the s1 baseline in length-controlled reasoning, generalization of this control to ood tasks.

**Questions To Authors:**

- How would LCPO compare with SFT/DPO methods with similar length control? Like [1] did?

Reference

1. Weizhe Yuan et al. Following length constraints in instructions, 2024.

**Reasons To Accept:**

- The proposed method, LCPO, is simple and effective in controlling cot length, which is important in achieving efficient reasoning.
- Experiments are comprehensive, especially the Appendices are abundant.

**Reasons To Reject:**

I don't find any major problems with this paper. Below are some minor ones.

- Presentation can be improved.
	- There are repetations in Sec. 4, (Models and Datasets & Evaluation and Implementation Details.)
	- Details are missing. How do you implement s1 on DeepScaleR?
- I don't understand Table 1. Why are L1-Exact and L1-Max listed 3 times with different results? Seems they are different runs with different length control. But there are no further demonstrations, and I think it would be more proper to pre-determine the length than pick results afterwards.

---

> ### Author Response · Authors · 2025-06-03
>
> We thank the reviewer for their thoughtful review and positive feedback regarding the simplicity and effectiveness of LCPO, and our extensive experimental evaluation. We are also glad that the reviewer found no major problems. We address the reviewer’s minor concerns below.
>
> ---
>
>
> > There are repetations in Sec. 4, (Models and Datasets & Evaluation and Implementation Details.)
>
> Thank you for pointing this out. We will update the section to eliminate redundancies.
>
> ---
>
>
> > Details are missing. How do you implement s1 on DeepScaleR?
>
>
> S1 is a method that forces the reasoning to end by inserting a special </think> token once the maximum tokens is reached and then prompts with “Final Answer.” The exact code used can be found [https://anonymous.4open.science/r/L1-COLM-7832/scripts/baselines/eval_s1.py](here). We follow exactly the same procedure as described by S1’s authors and will explicitly detail this implementation in the camera-ready.
>
> ---
>
>
>  > I don't understand Table 1. Why are L1-Exact and L1-Max listed 3 times with different results? Seems they are different runs with different length control. But there are no further demonstrations, and I think it would be more proper to pre-determine the length than pick results afterwards.
>
> Our goal is to compare our models against state-of-the-art models at comparable token lengths. We evaluated multiple (3) baseline models, noted their token lengths, and for each length we prompted the L1-Exact model to generate with the same token length. Therefore L1-Exact and L1-Max are listed three times. We will clarify this better in the revised paper.
>
> ---
>
>
> > How would LCPO compare with SFT/DPO methods with similar length control? Like [1] did?
>
> SFT/DPO, as used in previous works [1], would be ineffective for length control in reasoning models. This is because they rely on generating seed data containing outputs of different token lengths, and relabelling them based on the actual length of the generated response. However, as we show in Table 5 (Appendix B), without training, reasoning models do not follow the length constraint and have a narrow range of output lengths. As a result, offline methods such as SFT/DPO would be ineffective. In contrast, because LCPO rewards are based on closeness to stated length, the narrow distribution of length gradually increases, until following with high precision. Here is the Table 5 for reference:
> | Requested Length | Generated Length |
> |------------------|------------------|
> | 512              | 5491             |
> | 1024             | 6072             |
> | 2048             | 5846             |
> | 3600             | 6421             |
>
> ---
>
> Thank you again for your insightful comments; please let us know if there are further points we can clarify.

---

> > ### Comment · Reviewer_ivGY · 2025-06-04
> >
> > Thanks for your feedback and I think it is a good paper.
> >
> > For the last question, we can always generate seed data without length control, then count their tokens and put them back into the prompt. By this it is possible to construct SFT/DPO data with length requirements and corresponding responses.

---

> > ### Author Response · Authors · 2025-06-11
> >
> > We are glad the reviewer found the paper good!
> >
> > Regarding SFT data, we followed the methodology as suggested by the reviewer. The results are shown in the following table:
> >
> > | Requested Tokens | Actual Tokens |
> > |------------------|---------------|
> > | 512              | 21388         |
> > | 1024             | 22749         |
> > | 2048             | 21426         |
> > | 4096             | 20903         |
> >
> > In essence, we find that using SFT alone, the model *doesn't learn to follow length constraints* in the prompt. This is primarily due to two factors: 1. the small distribution of output lengths for a given question, causing the model trained on such data to ignore the length constraint, and 2. inability to follow length constraints in prompts, even to the slightest extent. We further elaborate on the experimental setup along with detailed reasoning for the poor performance of SFT [here](https://anonymous.4open.science/r/L1-COLM-7832/results_sft/README.md). We will include the discussion in camera-ready

---

### Official Review · Reviewer_wXkP · 2025-05-13

**Rating:** 8
**Confidence:** 4
**Ethics Flag:** 1

**Summary:**

The length of chain-of-thought reasoning in reasoning language models is typically not bounded,  making it impossible to allocate test-time compute to achieve a desired level of performance. The paper proposes Length Controlled Policy Optimization (LCPO) to optimize for accuracy and adherence to user-specified length constraints via reinforcement learning.
Experiments show that LCPO outperforms previous test-time scaling methods.

**Reasons To Accept:**

- The paper tackles an important problem that reasoning language models often underthink or overthink, resulting in performance degradation and waste of compute.

- The paper proposes a simple yet effective method called length controlled policy optimization, which optimizes a pre-trained reasoning language model via reinforcement learning to follow the instructions on response lengths.

- Experiments are comprehensive, demonstrating that the trained models outperform other length-controlled models while maintaining strong performance and can generalize effectively to out-of-domain tasks.

-	The paper is well-structured and clearly written.

**Reasons To Reject:**

I don’t see any major weaknesses in this paper.

---

> ### Author Response · Authors · 2025-06-03
>
> We thank the reviewer for their thoughtful review and appreciation of the importance and effectiveness of LCPO. We particularly appreciate your recognition of our comprehensive experimental setup and clarity in writing. We are delighted you found no major weaknesses.

---

> > ### Comment · Reviewer_wXkP · 2025-06-10
> >
> > Thank you for your response. I will maintain my original rating.

---

### Author Response · Authors · 2025-06-11

We thank all the reviewers for their thoughtful reviews and discussions! We are glad that all the reviewers found that our work tackles an important and timely problem, and appreciated the strong results of L1 along with comprehensive evaluation and analysis. Further, during the discussion period, we added two new results:

1.) We show that methods such as direct supervised finetuning is ineffective, highlighting the need for LCPO in length-control for reasoning models, [Link](https://anonymous.4open.science/r/L1-COLM-7832/results_sft/README.md) \
2.) We scaled the LCPO recipe to a larger 7B reasoning model, showing high controllability and much stronger performance than baselines. These results highlight the effectiveness and potential of LCPO to scale to even larger reasoning models. [Link](https://anonymous.4open.science/r/L1-COLM-7832/results_7b/README.md)

We once again thank the reviewers for the active discussion, which helped us strengthen the paper even further!

---

### Decision · Program_Chairs · 2025-07-08

**Decision:**

Accept

**Comment:**

The paper proposes LCPO: a simple RL technique for controlling the output length of an LLM while optimizing for accuracy. The reviewers commend the paper for the effective technique, comprehensive experiments, and subsequent analysis on the new "L1" model, including a finding that tuned small models with short CoT can outperform larger models. Most reviewers pointed out that the paper draft, as written, was missing details or had other presentation issues (such as the concerns around Table 1), which the authors have agreed to fix. Otherwise, the reviewers were positive and therefore I recommend acceptance.